# Secure Z-MAC Protocol as a Proposed Solution for Improving Security in WSNs

**Mona Nasser Almansoori** [1] , **Ahmad Ahmad Elshamy** [1] **and Ahmad Abdel Muttalib Mustafa** [2,*]

1 Department of Mathematical, Faculty of Science, Computer Science Unit, Suez Canal University, Suez, Ismailia 43711, Egypt; mnmalmansoori@gmail.com (M.N.A.); ahmed_alshamy@science.suez.edu.eg (A.A.E.)
2 Faculty of Information System and Computer Science Unit, British University, Cairo 11837, Egypt
* Correspondence: ahmad.mostafa@bue.edu.eg; Tel.: +971-505-686-660

**Abstract:** Security is one of the major issues in Wireless Sensor Networks (WSNs), as poor security disrupts the entire network and can have a significant effect on data transmission. WSNs need safe data transmission at a high rate while maintaining data integrity. By modifying the Z-MAC protocol and merging it with IHOP and elliptic-curve encryption techniques, the present research produced a novel protocol that enables safe data transfer. Additionally, the paper examined the IHOP technique for secure data transfer based on the Z-MAC protocol, which offers a simple and efficient key generation mechanism based on a hierarchical key management architecture. Additionally, the Z-MAC protocol offered low contention, high throughput, reduced latency, low power consumption, and increased efficiency. One of the most major applications of the Secure Z-MAC protocol may be the Vehicle Area Network, which would help in increasing highway automobile traffic while simultaneously enhancing individual safety and minimizing accidents.

**Keywords:** security in WSNs; security attacks; Z-MAC protocol; secure Z-MAC protocol





## 1. Introduction

A wireless sensor network (WSN) is a set of nodes that cooperate to build networks. Every node contains processing capabilities, several kinds of storage (data memory or program memory), a transmitter or a receiver, a power source (such as a battery or photovoltaic panels), and numerous sensor systems or independent sensors. The nodes interact remotely and without a wired connection, and they often organize after they are deployed in the network [1]. IEEE 802.15.4 is mainly utilized for short-range sensor networks, and one of the protocol's main aspects is that its MAC layer continues to support contention-based and contention-free access (schedule-based) [2]. Contention-based protocols are mainly used for local networks, and two primary examples are the Timeout MAC (T-MAC) and Berkeley MAC (B-MAC) protocols. Contention-free access protocols, also known as schedule-based protocols, work based on a time-division technique, and one example of this type of protocol is the Data-Gathering MAC (D-MAC) protocol. In addition, there are hybrid protocols, which are a combination of the carrier-sense multiple access (CSMA) and time-division multiple access (TDMA) systems; an example of this is the Z-MAC protocol.

The Zebra MAC (Z-MAC) protocol is created by merging the characteristics of the CSMA and TDMA systems. In a wireless network configuration, Z-MAC integrates specific times and packets to provide contention-free channel accessibility [3]. The nodes are also permitted to utilize slots that they do not own by using the CSMA method and prioritizing back-off time periods. To summarize, Z-MAC employs the CSMA method when there is little traffic and the TDMA technique whenever there is a lot of traffic in the network. Under low traffic volumes, Z-MAC operates identically to CSMA [3]. During times of heavy traffic, competition for slots is fierce. Z-MAC requires an explicit setup step (which takes time and energy), whereas explicit contention notification (ECN) messages may be utilized to relieve

localized contention [3]. When there is significant contention in the network while utilizing the CSMA method, the Z-MAC protocol enables sender nodes to send more data, which improves time synchronization and protocol efficiency, and is comparable to the TDMA strategy. As a result, there is less latency, the quality of the information sent is excellent, and there are fewer delays while sending data packets. It also allows for overcoming the issue of the hidden terminal problem, which contributes to enhancing the network and data security.

The Z-MAC protocol also enables having minimal contention, fast throughput, shorter latency, lower power consumption, and improved efficiency [3]. Because of weak data integrity and encryption methods, wireless sensor networks are readily hacked. To enable real-time data accessibility, a lightweight encryption method is needed to protect wireless nodes and the data generated by these nodes. Hence, the current research has focused on increasing the security of WSNs by developing a more secure and safe Z-MAC protocol. Also, some of the contributions of the current research are discussed below. The current study will contribute to:

(1) Increasing the security of WSNs implemented in various networks, such as the transportation network, smart city network, and complex communication networks;
(2) Studies of future researchers when they are discussing implementing a new framework to increase security in WSNs and its applications;
(3) Safe data transfer with the encryption of packets using the elliptic curve's efficient and lightweight characteristics, which would led to attaining high throughput without compromising network security.

The following sections outline a critical review of the types of security attacks, the significance of security and encryption methods, and how the Z-MAC protocol operates in a wireless network, along with its benefits and limitations.

## 2. Literature Review

### 2.1. Types of Security Attacks in Networks

Security networks are highly vulnerable to attacks, which allows hackers to modify data, hinder data transmission, steal information, attack the confidentiality of individuals, and other such issues [4]. Because sensor networks are scattered, tracking and regulating the true state of the network's elements is a difficult task [5,6]. Any fault in any of its elements may go unnoticed, or the fundamental source of the problem could be unknown. There are various types of attacks that an attacker can engage in to create issues in a network. Attacks may be classified as either external or internal attacks [7]. An attacker from the outside does not know the bulk of the encoded data in WSNs, whereas an attacker from the inside might have partial knowledge of the key and/or about other sensor networks [7]. Internal attacks are considerably harder to detect and combat [5].

Some of the most common attacks that occur in a WSN are eavesdropping attacks, sinkhole attacks, denial of service (DoS) attacks, flooding attacks, and blackhole attacks, among many others [8–13] The description and classification of these attacks are described in Table 1.

Also, there are various technological variants that can help in avoiding the above-mentioned attacks in WSNs, such as blockchain technology, trust-based systems, and cryptography, among other such technologies. For instance, the authors of [13,14] discussed how blockchain technology can strengthen data transfer and security in WSNs. The authors mention that a blockchain's primary benefit of decentralization allows data storage on multiple servers, which also contributes to high security in a network. One other technology discussed by [15,16] is cryptography. One of the primary issues of the WSN is its implementation, for which cryptographic techniques can be used. One such technique is the elliptic-curve encryption technique [15], which is used in this research and discussed in detail in Section 2.3. An additional technique that can prevent security attacks in WSNs is a trust-based system or trust management system [17,18]. The authors further explain the benefit of the trust management system in WSNs, which explains that trust management

models make decisions for enhancing the security of WSNs based on the trust values for nodes.

**Table 1.** Types of attacks in a wireless sensor network.

| Attack | Type of Attack | Description |
| --- | --- | --- |
| Eavesdropping attack | External and internal | The action of an illegal party intercepting information and data is known as eavesdropping. Because of the dispersed nature of the broadcast channel, any attacker may capture messages at a selected frequency and obtain confidential data about a system's operational processes, as well as see physical characteristics by employing a sensor node [8]. |
| Sinkhole attack | External and internal | A sinkhole attack occurs when an attacker or hostile sensor node disguises itself as a sinkhole in such a way as to entice and collect all traffic in a sensor network. An attacker listens in on-demand patterns, then presents these to the targeted system as having the best efficiency or being the quickest path to the base station. By placing itself between cooperative nodes, the intruder is likely to modify the data passing between them and launch an attack, rendering the WSN very susceptible [9]. |
| DoS attack | External and internal | By partially or fully stopping the activities of a server connected to the Internet, an attacker tries to make a computer, networking device, or program unavailable to its intended clients. A DoS attack is often carried out by bombarding the targeted computer or services with numerous queries in an attempt to overload networks and prevent some or all legitimate requests from being fulfilled [10]. |
| Flooding attack | External and internal | The flooding attack allows intruders to possibly send a huge amount of traffic to particular servers or applications to deplete all of their resources responding to false traffic, leaving them unable to fulfill legitimate service requests [11]. |
| Blackhole attack | External and internal | An unauthenticated node that is likewise an eavesdropper serves as a blackhole, monitoring for request packet communications from its neighbors and responding with incorrect and misleading information about the shortest route to the sink node [12]. |

These attacks can harm the security of the networks, so it is important to outline the significance of security in sensor networks, which is carried out in the following section.

*2.2. Significance of Security in Sensor Networks*

Managing security in sensor networks contributes to authenticity, secrecy, integrity, privacy, and access to data packets and information. When additional security elements are introduced into the network, the computations, communications, and administration overheads increase in tandem with the increased security effectiveness [19]. Secrecy, which is the fundamental objective of security, is one of the most difficult barriers to conquer to guarantee authenticity and access to information, as well as the accomplishment of time-critical and essential goals. Information security may prevent hostile nodes from gaining access to data; however, it cannot protect data from being changed by unapproved parties [20]. Data integrity guarantees that the message is not tampered with during transmission. A malicious node may lead the networks to malfunction by interfering with the message. Additionally, communications may be interrupted during transfer even when no hostile nodes are present. To guarantee data integrity, data encryption protocols must be used.

There is a need to study many situations in terms of increasing security in various contexts [21]. For instance, [22] have talked about maintaining security in healthcare WSNs, and [23] has discussed ensuring security in a WSN related to smart cities. Furthermore, [24] have focused, in detail, on maintaining security in WSNs linked to military applications.

Privacy needs vary by context, and WSNs may be used in a wide variety of situations. When WSNs are coupled with IoT technology, both the IoT's [21] and WSN's security needs must be addressed. This implies that cryptographic protocols must also be addressed in light of the unique context wherein WSNs may be employed. For instance, in a hospital setting, a greater degree of confidentiality is necessary to ensure the security of clinical information [21]. However, for these particular cases, fundamental security standards such as data secrecy and integrity might be enforced. For instance, while analyzing the security needs of a particular cloud computing system that employs WSNs, fundamental security criteria should be addressed. Even though the public cloud and similar situations are not detailed, the security standards and remedies are applicable if WSNs are employed in a particular context.

### 2.3. Elliptic-Curve Encryption and the IHOP Mechanism for Key Generation

Elliptic-curve encryption (ECE) is a kind of encryption method that may be used for encoding and decoding data, as well as for message authentication, developing digital signatures, and engaging in the process of exchanging keys [25]. Keys in ECE are much smaller in size than keys in other cryptographic algorithms such as RSA. A smaller key means simpler data processing and information management, fewer technical needs (with regards to buffering and data storage), reduced bandwidth when sending keys across networks, and a greater battery capacity [26]. The elliptic algorithm encrypts data by multiplying elliptic-curve points. Elliptic-curve point multiplication is the process of continuously adding and growing the numerical values and the elliptic curve in an attempt to encrypt and secure particular data [27,28].

While executing the encryption procedure described below, the elliptic-curve encryption method utilizes an elliptic curve in conjunction with a finite field.

$$E: y^2 = x^3 + ax + b \tag{1}$$

The following Equation (1) outlines the plain and primary curve of an elliptic curve, in which:

'E' represents the elliptic curve;

'a' and 'b' represent rational numbers or integers;

'x' and 'y' represent rational numbers in the primary elliptic curve.

The specified elliptic curve has also been processed in the Abelian group, which is primarily taken from the divisor group and is represented as follows:

As stated in the following Equation (2), the Picard group of E is represented by Pic (E).

$$D (E) \rightarrow P (E) \tag{2}$$

where, D (E)—Divisor of Elliptic Curve

P (E)—Element of Picard Group

Considering '0' degree divisors on an elliptic curve E, the following equation can be represented:

$$D^0 (E) \rightarrow P^0 (E) = E \tag{3}$$

From the following Equation (3), each of the variables is represented as follows:

$D^0$ (E)—'0' degree divisors on an elliptic curve E

$P^0$ (E)—'0' degree divisors on Picard group of E

There are different kinds of encryption algorithms, such as the elliptic-curve digital-signature algorithm (ECDSA) and elliptic-curve Diffie–Hellman (ECDH), that are implemented while carrying out a transaction.

The elliptic algorithm performs encryption operations by multiplying elliptic-curve points. Elliptic-curve-point multiplication is the technique of continually adding and increasing the point value and the elliptic curve in order to maintain the protection and

security of specific data (Kang et al., 2018). Scalar multiplying occurs at this level, which is denoted by the Hessian Elliptic Curve (HEC).

'E' represents the elliptic curve that consist of some finite fields, and the point multiplication of these points is represented as follows:

$$nA = A + A + A + A + A + \ldots + A \tag{4}$$

The variables used in Equation (4) are represented as follows:

n—integer

A—a point that lies on an elliptic curve

The Logarithmic function may be determined by the level of n in the curve, which happens when it is greater than the other values (A) of the curve.

The basic process for understanding how ECE works is discussed as follows:

1.  A network's nodes decide on an elliptic curve and a fixed curve point F, which are not a secret in the network;
2.  Node A selects a secret random integer $A_k$, which is a secret key, and a curve point $A_P = A_k F$ is calculated, which is considered a public key;
3.  Node B also engages in a similar process as the one conducted by Node A;
4.  Node A randomly selects an integer $B_k$ and it is considered a secret key that is not to be shared publicly. After that, it calculates the curve point $B_P = B_k F$ as its public key;
5.  Once the secret key is generated, Node A wants to send a message to Node B;
6.  Node A can simply compute $A_k B_P$ and use the final developed outcome as the secret key for encrypting a conventional symmetric block;
7.  After this step, Node B can further calculate a similar value by evaluating $B_k A_P$, because

$$B_k A_P = B_k \cdot (A_k F) = A_k \cdot (B_k F) = A_k B_P \tag{5}$$

The way in which elliptic-curve encryption works is represented in the following Figure 1.

The security of the above-mentioned approach concerns the assumption that it is significantly challenging to calculate the value of k given F and kF, which would not allow the attacker to attack the network or system.

One other very crucial mechanism is the IHOP mechanism scheme, which is primarily used for data authentication. There are five main stages of the IHOP mechanism scheme, which are highlighted by [28]:

1.  The initial stage is installation and the setting-up stage for nodes. In this stage, the main server provides each node with a distinctive and different identification code, and provides them with the keying resources needed to establish paired keys with some other nodes. After setup, a node generates a one-hop paired key with each of its neighbors.
2.  The second stage is the association discovery stage, which has two further stages known as the base-station hello stage and the cluster acknowledgment stage. In the base-station hello stage, the base station sends a HELLO message to all the nodes; this allows them to store the identification codes of the nearby t + 1 nodes, which are on the path towards the base station. In the cluster acknowledgment stage, the cluster head of the network sends an acknowledgment to the base station.
3.  The third step is the report endorsement phase, in which t + 1 nodes cooperate to generate a report when they detect the existence of an event occurrence. To be more specific, each participating node computes two MACs over the event—one by the key supplied by the base station and the other employing the bilateral key shared with the nodes above. The MAC addresses are then sent to the cluster head. The cluster head collects MAC addresses from all the nodes in the network, combines them into a transmission message, and transmits it to the base station.

4.　The fourth stage, en-route filtering, involves each sending node validating the message authentication code (MAC) produced by its subordinate associated and connected nodes, before excluding it from the receiving report. Following successful verification, the node creates and deploys a new MAC, with the bilateral key exchanged with the upper connected node. Lastly, it sends the information to its next node throughout the route of the base station.

5.　Following the receipt of the message, the base station validates it in the last stage, known as the base-station verification stage. If the base station determines that the communication was properly authorized by the t + 1 nodes, it acknowledges the message; otherwise, it dismisses it.

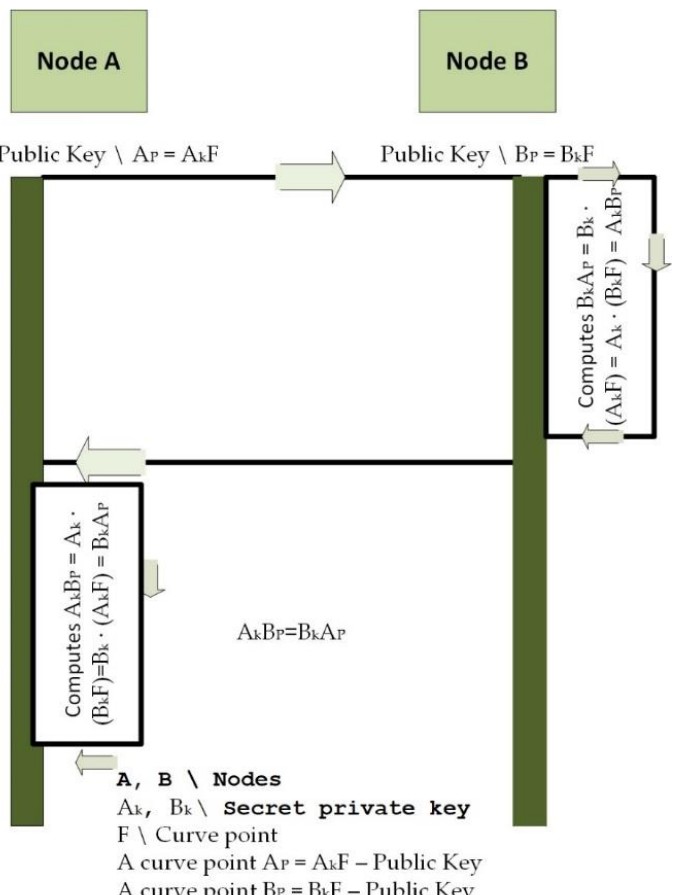

**Figure 1.** Elliptic-curve encryption.

　　The representation of the IHOP key-sharing mechanism is shown in the Figure 2. However, before understanding the key-sharing mechanism using IHOP, it is important to understand the IHOP mechanism alone, which is described in the Figure 3.

　　These encryption mechanisms would be deployed with the Z-MAC protocol to enhance its security. However, before discussing this, it is important to understand the basic workings of the Z-MAC protocol along with its benefits and limitations.

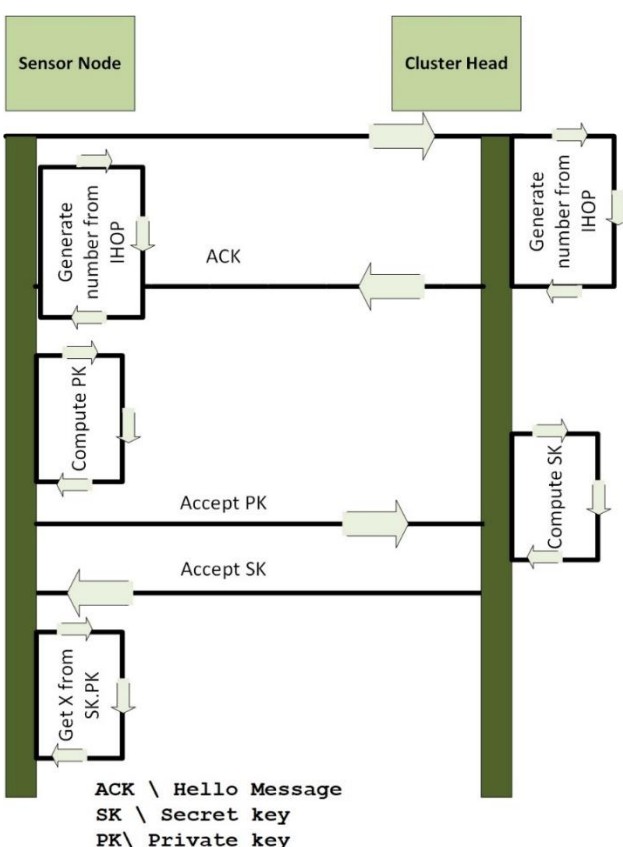

**Figure 2.** Key sharing using IHOP mechanisms.

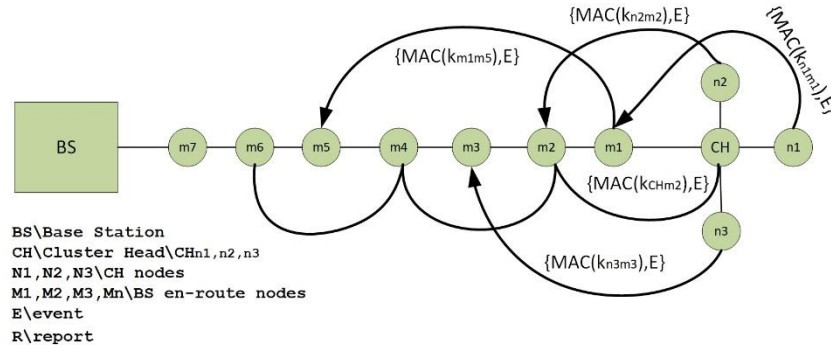

**Figure 3.** IHOP Mechanism.

## 2.4. Z-MAC Protocol

In [3,29], the authors suggested the Z-MAC protocol for WSNs. The authors of [3] use the distributed RAND (DRAND) method, which is a reliable and effective multi-channel scheduling solution. It is the first distributed version of the random method, a well-known yet logically centralized scheduling approach. It maintains an identical channel efficiency to the random approach; however, it has the text and running time complexity of $O(\delta)$ [30]. A basic description of how the Z-MAC protocol operates is represented in the Figure 4.

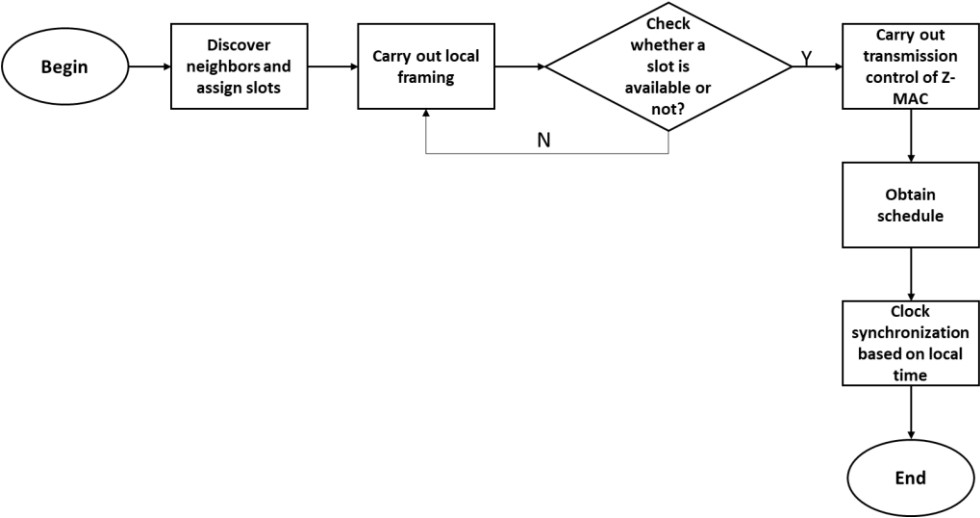

**Figure 4.** Z-MAC Protocol.

When CSMA and TDMA are coupled, Z-MAC becomes more resilient to timing problems, time-varying connectivity circumstances, slot assignment errors, and interruptions; in the worst-case scenario, it behaves similarly to CSMA. Furthermore, Z-MAC is resistant to changing topologies and has clock synchronization issues, both of which are prevalent in WSNs [30]. The Z-MAC design concept is that the high initial expenditures are recovered throughout the network's lifespan, ultimately being compensated by increased capacity and energy effectiveness. Z-MAC aims to minimize intrusions during intervals when users have packets of data to transmit, and to give them early opportunities to send them all while pre-scheduling the slots to prevent collisions [30].

However, there are some limitations in the originally developed Z-MAC protocol, such as issues with regards to hidden terminal problems, poor synchronization between the sender and receiver, high use of energy, and other such issues. Due to these issues, security problems also occur, especially due to hidden terminal problems and poor synchronization issues [30]. Hence, the current research has focused on developing a secure Z-MAC protocol, which would enhance the security of the sensor networks.

## 3. Proposed Solution—Secure Z-MAC

### 3.1. Secure Z-MAC Protocol

A secure Z-MAC protocol would occur in seven developed stages, such as neighbor discovery, slot assignment, local framing, transmission control, explicit contention notification, the receiving schedule of Z-MAC, and local time synchronization:

1.  The step of neighbor discovery starts with the use of a neighborhood detection algorithm. The present method monitors at 30 s at irregular intervals. The network nodes that store data about one-hop neighbors, acquired through ping requests and messages, verify nodes within two hops. The energy used by DRAND is proportionate to the time when the transmission schedule is given. An attacker may impersonate a genuine node by using node replication. A new intruder, for example, might connect to the network as a neighbor and transmit incorrect data. Therefore, hierarchical keys are created by distributing the primary key and shared keys among n + 1 nodes.

2.  In the local framing stage, each node can retain a local time allotment that is proportional to the extent of its neighborhood, while remaining strategically far from any dispute with competing neighbors. If node A is in a two-hop neighborhood but uses a time slot granted by node B for whatever reason, the effect will last until the nth slot. The transmission rule enables DRAND to assign nodes to the nth time frame within the anticipated time range. Once a node achieves this level, it is only allowed to utilize its very own slots once every six seconds under any circumstance. In the

Z-MAC protocol, CSMA is permitted to access empty slots to guarantee that they are completely used. The process of local framing is presented in the Figure 5.

**Figure 5.** Local framing.

1.  In the transmission control stage, during the startup, the Z-MAC algorithm has a fixed slot allocation. A node is considered to exist in High Contention Level (HCL) mode when it obtains an explicit contention notice (ECN) in the nth time frame. When sending in Low Contention Level (LCL) mode, each node may select any time frame to send the packet; however, only the node's one-hop neighbor and owner are permitted to send. This will help to build trust among nodes, and also generate safe, secure, and encrypted private keys. When a node is prepared to send data in the nth slot, it is verified to check whether it holds the slot. If this is not the case, the data are transmitted when the slot is operational. If the connection is free, the node takes the slot; otherwise, it waits until the channels are available. Even though a node has additional time, an interchange will be rejected if the broadcast does not finish within the timespan of the availability.

2.  In the ECN stage, the ECN sent to the nodes about the LCL or HCL condition for slot assignment. After each schedule, the node sends the ECN notifications to check whether any other nodes are competing for a similar data transmission slot. When node-to-node contention in slots rises, the channel's size generally expands as well. MICA2 was used to examine interactions between the transmitters and the two-hop neighbor of the sender.

3.  The fifth stage is receiving the schedule of Z-MAC. Although the Z-MAC and B-MAC protocols approach energy consumption differently, both are identified as idle monitoring devices during periods of low movement. Because a node in the HCL communicates exclusively between two specified areas, a high slot size may lead to considerable latency. When there is minimal contention, storage has little impact on transmission delay, since nodes are constantly preparing to transmit data. There is a strong connection between system latency and slot size. As a result, a big slot size may result in an inordinate amount of delay.

4.  In the local time synchronization stage, periodic synchronization is required for the Z-MAC protocol to run the HCL in the case of several collisions. This method allows

passive receivers to sync their clocks with the data sent by the transmitter without requiring any synced communication. This Z-MAC method would be employed to synchronize the clock timings of every node and calculate a weighted moving average among them, allowing for the preservation of a trust factor. As a result, the receiving nodes will know which one to trust, as a new node will be unaware of the time gap between the nodes. Due to the extra packets sent and received by the node that exchanges synchronization signals in this network, the node that sends and accepts synchronized signals often has a higher trust element than the other nodes.

The following formulas are used to synchronize the time schedules between the two nodes.

$$C_{avg} = (1 - \beta_t) * C_{avg} + \beta_t * C_{avg} \tag{6}$$

$C_{new}$—New clock time shared between neighboring nodes
$C_{avg}$—Weighted average clock time calculated
$\beta_t$—Synchronization rate at which a node should transmit clock time
A pictorial representation of the above-discussed secure Z-MAC protocol is carshown in the Figure 6.

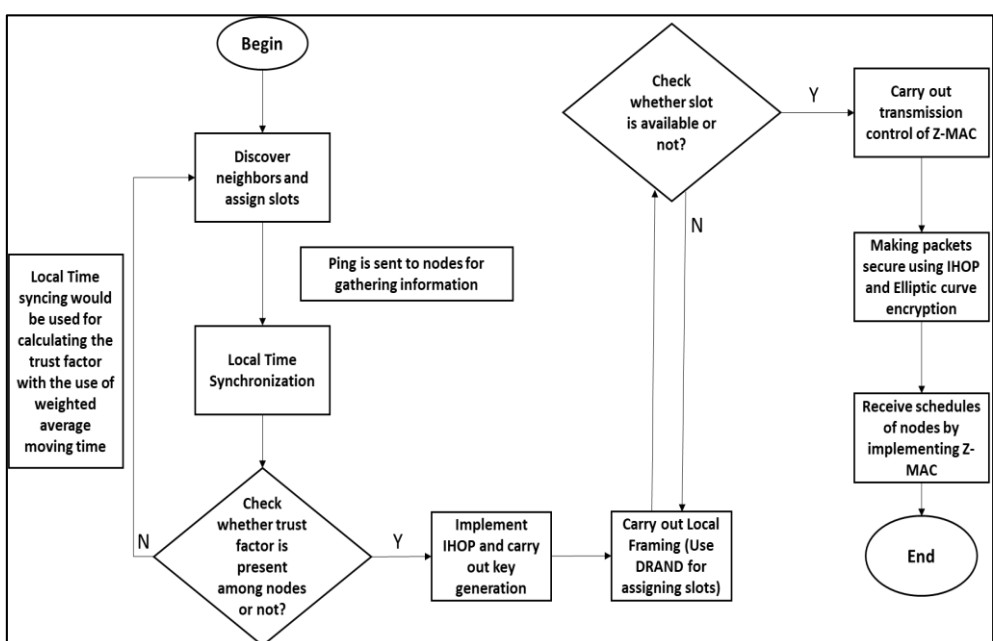

**Figure 6.** Secure Z-MAC protocol.

The above-presented figure is the secure Z-MAC protocol. The steps of the Z-MAC protocol are made more secure by incorporating security-based mechanisms in the original protocol. Additionally, the Z-MAC protocol's secure layer is optimized for speed and the use of a compact algorithm. Although it provides strong encryption, we found that an elliptic curve is extremely effective in cases when robust encryption is required without sacrificing efficiency. To generate and exchange keys effectively and securely, the IHOP technique was utilized. Additionally, this technique was utilized to detect and report sensor network attacks and breaches. In the current study, a similar concept was used, but the method was linked with Z-MAC operations to ensure the security of a WSN's function. The primary focus was on enhancing the security among nodes by focusing primarily on making the key-exchange process safe and secure. Additionally, Z-MAC includes a mechanism for locating neighbors to start node-level data sharing. As a consequence, a secret key that was network-independent is required. IHOP works by performing an XOR function on the transmitting nodes' MAC addresses. A similar concept was employed in the current study, whereby an XOR relationship on the node's MAC address was implemented to create a

128-bit key among nodes, which was exchanged solely among neighboring nodes. This method allows for the construction and management of hierarchical keys, which is very effective for managing security in the network. Once keys were transferred between the nodes of the network, it was essential to build credibility among them and to create keys between nodes to avoid node replication attacks. The current study examined the concept of trust via the application of Z-MAC clock-synchronization time-framing techniques. This helped in the development of a list that can be depended upon for information sharing inside a network. The accompanying graphic XX depicts the whole process of Z-MAC operations that are conducted to enhance the security of the network.

### 3.2. Secure Z-MAC Algorithm

The secure Z-MAC algorithm (Algorithm 1) is presented as follows, which provides a step-by-step understanding of the secure Z-MAC.

| **Algorithm 1** Secure Z-MAC Algorithm |
|---|
| Step 1: Begin |
| Step 2: Conduct the neighbor-discovery and slot-assignment process |
| Step 3: Send a ping to collect details about the neighboring nodes and to perform local clock synchronization |
| Step 4: Check if there is a trust factor between the neighboring nodes |
| **IF** 'Yes' **then** implement the IHOP mechanism and generate keys |
| **IF** 'No' **then** repeat Step 2 |
| Step 5: Implement the local framing stage |
| Step 6: Check if there is a slot available |
| **IF** 'Yes' **then** carry out the transmission control of the Z-MAC protocol |
| **IF** 'No' **then** repeat Step 5 |
| Step 7: Carry out the encryption of packets using the IHOP mechanism, which is run by ECE |
| Step 8: Receive schedule of the Z-MAC protocol |
| Step 9: End |

## 4. Research Methodology and Security Analysis of the Developed Protocol

In the current research, nodes were designed to be at an equal distance from each other, and the main collector nodes were put in to ensure that for similar packets, the same bandwidth was utilized. Before transmitting the data, we made sure that there was no middle node in between, and that each node had one neighboring node. This technique was used to ensure that each node had similar throughput as they were strategically placed in the same setup, and that there were no delays recorded as all the nodes were placed roughly 2 m apart; this helped us in improving the performance of the modified Z-MAC with the new functions that are proposed in the paper.

The performance evaluation of the secure Z-MAC Protocol was also carried out using an NS2 simulation tool. There are many methods that can be used for analysis to ensure the security of protocols, such as program analysis (code analysis), statistical analysis, software test analysis, model extraction methods, execution trace analysis, and net trace analysis [31]. The current research used code analysis and statistical analysis to conduct the security analysis of the Secure Z-MAC protocol. The parameters and values considered in the research are represented in the Table 2.

**Table 2.** Parameters and their Respective Values.

| Parameters | Values |
| --- | --- |
| Number of nodes | 5, 10, 15, 20, 25, 30, and 35 |
| Area size | $25 \times 25$ |
| Wireless bandwidth (In Mbps) | 2 |
| Simulation duration (bit/sec) | 1000–3000 |
| Packet size | 20,000–120,000 |
| Number of CH | 1 |
| Types of attacks discussed | 3 |
| Number of attack nodes | $1,2,3,\ldots,n+1$ |

Also, the code analysis was a primary analysis technique, which helped in ensuring the security of the protocol. Many researchers have used code analysis to ensure the security of their developed protocol, which has further increased the security of WSNs. Numerous researchers have tried to encrypt and secure the WSN using a variety of different approaches. However, before discussing the security analysis of the protocol, it is important to discuss various methods of increasing security in the network. For instance, ref. [32] used the Hamming Residue Method (HRM) to protect the developed network against fraudulent attacks. At each additional node, the HRM technique generated a new passcode. For a large set of potential attacking nodes, the strategy proved to be successful and straightforward. The work in [33] suggested a compelling, genuine method and encoding in approach for WSNs in IoT systems. However, owing to the evolving environment of networking systems, the strategy is less efficient given the difficulty of the distributing keys (mostly private keys). Ref [34] suggests a practical and cost-effective technique for clustered sensor networks based on hash chains and elliptic-curve Diffie–Hellman (ECDH) key exchange that enhances malevolent activity in a WSN. The suggested approach is more secure and always removes weaknesses inherent in current wireless sensor-node techniques. However, the approach lacked a method for key creation and sharing, which would have ensured the system's robustness.

The code analysis used in the current study to ensure the security of the developed secure Z-MAC protocol was performed by developing a C++ language code for the protocol. The technique described in this study presents the benefits of adopting the elliptical curve, which permits the adoption of smaller keys to achieve a comparable degree of security to existing algorithms. The IHOP system ensures key creation and sharing, allowing the base station to detect any unauthorized assault. Additionally, an XOR was integrated into the modified Z-MAC to ensure that the computing process is as efficient as possible. Also, a basic code analysis to ensure the security of the protocol used code based on a hierarchical key-generation technique, which is presented as follows:

The current research built a secure Z-MAC scheme on the basis of hierarchical identity-based cryptography, because of the uniqueness of the MAC address for each sensor node using IHOP; this helped in generation of shared keys between neighbor nodes when the neighbor discovery was is initiated by Z-MAC. The following section presents the results, which represent the findings and experimental setup of the current research.

## 5. Results

### 5.1. Experimental Setup

NS2 is a highly popular and extensively used tool for modeling small- and big-area networks, as well as local- and wide-area networks. It supports protocols, such as the TCP and the User Datagram Protocol (UDP); network origin behaviors such as the File Transfer Protocol (FTP) and Teletype Network Protocol (Telnet); and constant bitrate and variable bitrate (VBR) simulations. Additionally, NS2 supports multicasting and certain MAC layer protocols. The simulator may assist researchers in evaluating the functioning of their newly suggested and innovative ideas for research disciplines. Due to its distinct characteristics, the simulations of the current research were carried out using the NS2 simulator.

Two scenarios were developed in order to better understand how the Z-MAC protocol works and how it may be useful in communication systems. The first example, a basic one, will help us understand how DRAND and Z-MAC function, whereas the following situations will be employed to assess safe Z-MAC execution under shifting data loads and potential network threats. The nodes were constant in both instances, because DRAND does not consider time slots allocated to specific nodes throughout random node deployment and configuration.

Each scenario was simulated to determine the throughput of the network in proportion to the number of packets delivered. The throughput of a network is defined as the time needed for a packet to transition successfully from node n to node n + 1. The primary objective was to quantify throughput by including three more phases into the process than are currently present in Z-MAC. The major goal of this study is to maintain performance comparable to that of the Z-MAC protocol when encryption and IHOP are employed, since both have been shown to be very effective at safeguarding packet and data transmission. Additionally, the present study focuses on enhancing the security of the Z-MAC protocol throughout the transmission and networking processes.

The secure Z-MAC protocol was tested against the three most prominent attacks, namely a flooding attack, a DDOS attack, and a blackhole attack. Three simulations were also run before testing the efficiency of the protocol against these attacks. The experimental results are discussed in the following section.

### 5.2. Experimental Results

The three simulations tested were simulation 1 (measuring the total transmission time using the Z-MAC scheme); simulation 2 (measuring the total transmission time during the attack vector's introduction against the packets transmitted in the Z-MAC scheme); and simulation 3 (measuring the transmission time while implementing the IHOP mechanism with the Z-MAC protocol). This simulation was conducted using NS2 tools, and three different simulations were carried out seven times. The results and findings have been presented in the following section.

### 5.2.1. First Simulation

To begin, in the first simulation, a network of five nodes was built to simulate traffic using the basic Z-MAC protocol or setup. Following this, the network nodes were increased in number by five each hour to assess the node design and time-framing technique. This procedure was repeated until 40 nodes were achieved, at which time the attack vectors were successfully implemented into the nodes, overloading or flooding the network, increasing node-to-node latency, and forcing the base station to reject packets.

Table 3 summarizes several configurations ranging from five to thirty-five nodes and their associated transmission time in seconds.

**Table 3.** Total transmission time for nodes using the Z-MAC scheme.

| Nodes | Packets/sec in Bytes | Time in bits/s |
|---|---|---|
| 5 | 20480 | 1361.78478 |
| 10 | 35840 | 1367.985821 |
| 15 | 51200 | 1379.645156 |
| 20 | 66560 | 1403.241674 |
| 25 | 81920 | 1443.686908 |
| 30 | 97280 | 1489.758152 |
| 35 | 112640 | 1576.497647 |

The X-axis indicates the number of packets in bytes, and the Y-axis represents the transmission time in seconds in Figure 7. As shown in the graph, there is a strong connection between the number of packets and the time required to transmit them across nodes, indicating that Z-MAC processing is efficient. We increased the number of nodes and

packets in each execution, which increased packet transmission time. For instance, when the number of packets was increased from 20,480 packets to 112,640 packets, transmission time increased significantly from 1361.7 s to 1576.5 s. The increase in transmission time reflects increased network latency, which may cause the base station to discard packets. However, when attack vectors were added into the network, performance suffered significantly, as described in the second simulation.

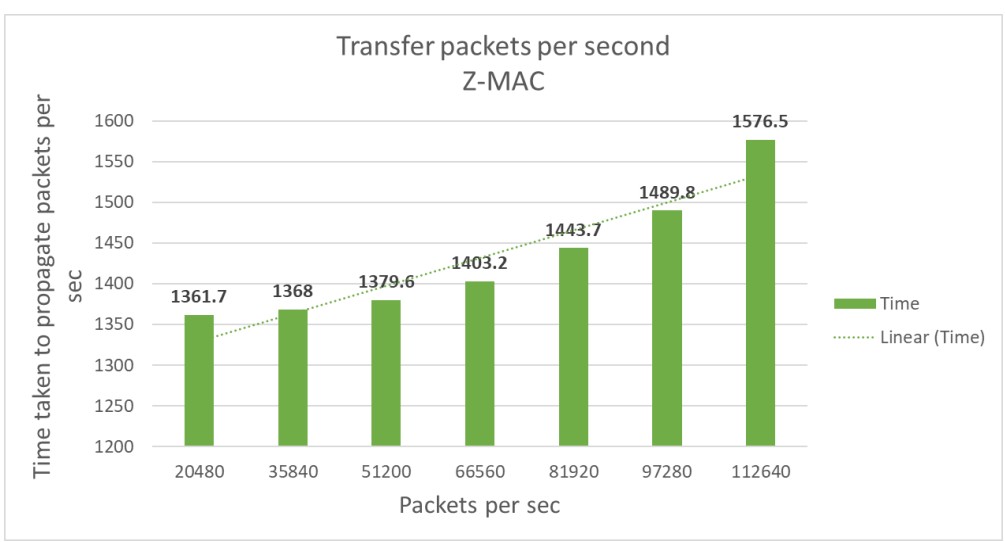

**Figure 7.** Transmission time using the basic Z-MAC protocol.

5.2.2. Second Simulation

The second simulation had a configuration ranging from five to thirty-five nodes. Additionally, an assessment of the Z-MAC operation was conducted in the present situation where a DDOS attack was inserted into the network. The flow configuration for the Z-MAC procedure was completed; however, after implementing the attack, its impact was evaluated using the setup's stages. The flow-setup time is the time required to complete six stages of the Z-MAC protocol, which are neighbor discovery, local framing, transmission control, ECN messages, receiving schedule, and local time synchronization, the results presented in Table 4.

**Table 4.** Total transmission time for nodes using the ZMAC scheme when an attack is introduced.

| Nodes | Packets/sec in Bytes | Time in bits/s |
|---|---|---|
| 5 | 20480 | 1357.1 |
| 10 | 35840 | 1359.5 |
| 15 | 51200 | 1579.645156 |
| 20 | 66560 | 1803.241674 |
| 25 | 81920 | 2072.686908 |
| 30 | 97280 | 2286.758152 |
| 35 | 112640 | 3725.497647 |

In Figure 8, the gradient in the graph reflects Z-MAC's efficient processing when an attack is introduced. Nevertheless, when the DDOS attack was added into the configuration, the time required for packet transmission increased, resulting in a delay in transferring packets from one node to another, and negatively impacting the system's performance. When the DDOS attack was conducted, an immediate increase in time consumed was noticed, affecting wireless network performance, and making them unusable for data-driven base station selection. For instance, in the beginning, when 20,480 packets were introduced, the transmission time was 1357.1 s; however, when the DDOS attack was

introduced later with the higher number of packets, which is 112,640, an abrupt increase in transmission time was seen, which was greater than the linear time indicating delay.

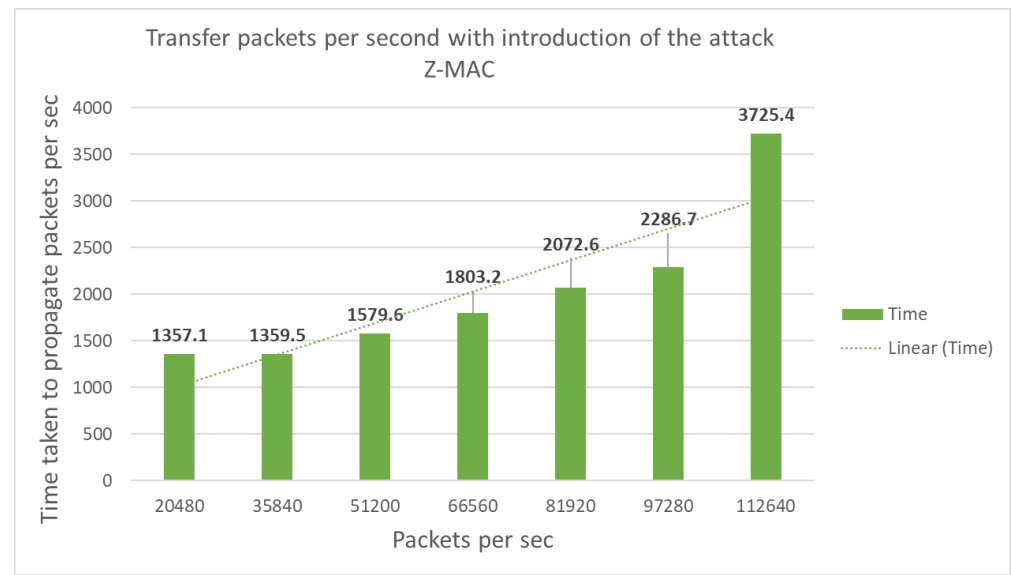

**Figure 8.** Total transmission time using the ZMAC scheme when an attack is introduced.

### 5.2.3. Third Simulation

After running two simulations, a third simulation was conducted. In the third simulation, the network's effectiveness was enhanced by including the IHOP mechanism alongside the key-sharing and generation processes, as well as by using the MAC address policy. Each node distributes the key at each phase through the XOR of the MAC, resulting in the blockchain's cipher text being utilized by the elliptic-curve method, which is represented in the Table 5.

**Table 5.** Total transmission time for nodes using the Z-MAC scheme along with the IHOP mechanism.

| Nodes | Packets/sec in Bytes | Time in bits/s |
|---|---|---|
| 5 | 20480 | 1358.598129 |
| 10 | 35840 | 1361.77138 |
| 15 | 51200 | 1366.963776 |
| 20 | 66560 | 1371.648392 |
| 25 | 81920 | 1375.062767 |
| 30 | 97280 | 1389.011017 |
| 35 | 112640 | 1409.034265 |

It can be observed that the Z-MAC protocol ran better when it was introduced along with the IHOP mechanism. Figure 9 also shows that the performance of Z-MAC is high compared to the first two simulations. However, when an attack is introduced in the network, it affects the transmission time.

To overcome this issue, an additional simulation was carried out with the implementation of the secure Z-MAC protocol. When the secure Z-MAC protocol was implemented, the transmission time increased; however, there was a delay in the transmission time. The results of the secure Z-MAC simulation are discussed in Table 6 and Figure 10.

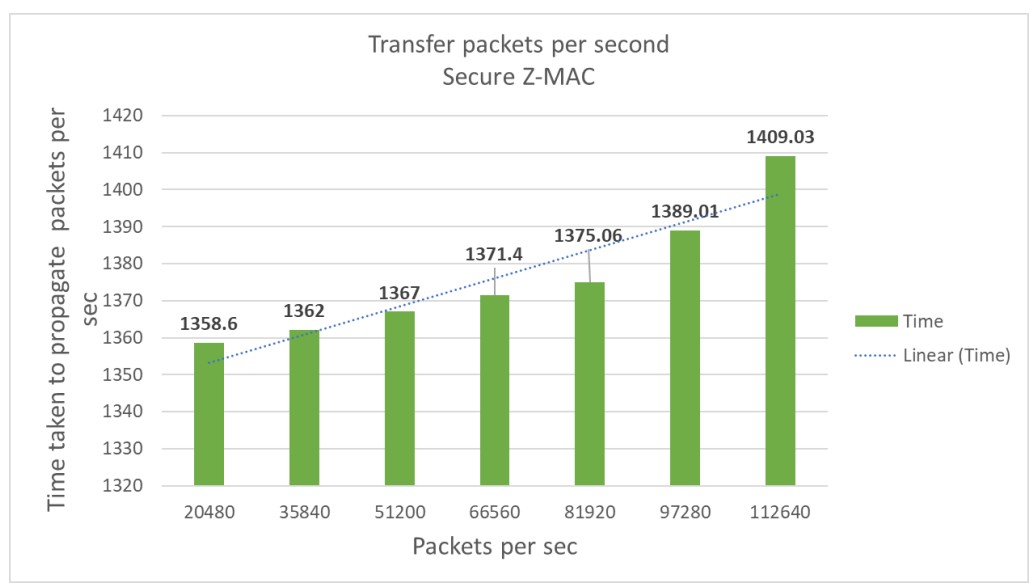

**Figure 9.** Total transmission time for nodes using the Z-MAC scheme along with the IHOP mechanism.

**Table 6.** Total transmission time for nodes using the secure Z-MAC scheme.

| Nodes | Packets/sec in Bytes | Time in bits/s |
| --- | --- | --- |
| 5 | 20480 | 1418.014112 |
| 10 | 35840 | 1464.402046 |
| 15 | 51200 | 1536.198699 |
| 20 | 66560 | 1645.619443 |
| 25 | 81920 | 1738.530628 |
| 30 | 97280 | 1833.294729 |
| 35 | 112640 | 1905.975264 |

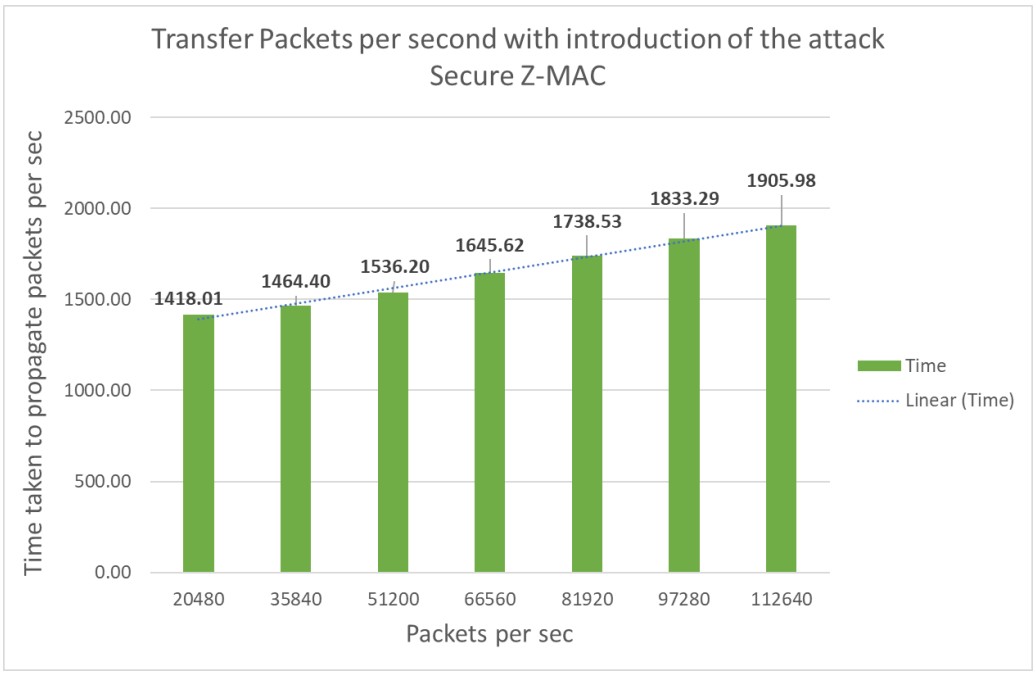

**Figure 10.** Total transmission time for nodes using the secure Z-MAC scheme.

If Figure 10 is compared to Figures 7–9, it can be observed that packet delay decreased significantly. Furthermore, the red line is comparatively more linear in Figure 10 than the other three simulations, which shows that the secure Z-MAC protocol is more efficient than the Z-MAC protocol. For instance, when only the Z-MAC protocol was being used, the delay was very high and there was a significant amount of difference in transmission time, which can be observed in Figures 8 and 9. However, with the use of the secure Z-MAC protocol, delays cannot be observed, and instead, there is a linear increase in the transmission time of packets. The delays in transmission time reduced significantly with the help of the secure Z-MAC protocol.

Furthermore, the efficiency of the secure Z-MAC protocol was ensured by introducing three types of attacks, such as flooding attacks, blackhole attacks, and DDOS attacks, which is represented in the Table 6.

### 5.2.4. Flooding Attack

The flooding attack was introduced to the network by flooding 60,000 packets in the network. The performance of Z-MAC and secure Z-MAC was compared in terms of transmission time, which is represented in the Table 7.

**Table 7.** Total transmission time during a flooding attack for nodes processed using the secure Z-MAC scheme compared to Z-MAC.

| Nodes | Packets/sec in Bytes | Z-MAC Transmission Time in bits/s | Secure Z-MAC Transmission Time in bits/s |
|-------|---------------------|-----------------------------------|------------------------------------------|
| 5 | 20480 | 135061 | 124567 |
| 10 | 35840 | 127404 | 1358.71 |
| 15 | 51200 | 183180 | 1360.58 |
| 20 | 66560 | 12678 | 1357.1 |
| 25 | 81920 | 114666 | 1356.7 |
| 30 | 97280 | 100000 | 1355.81 |
| 35 | 112640 | 195600 | 1383.11 |

In Figure 11, the X-axis shows the number of packets transmitted over the network in various configurations of nodes, whereas the Y-axis represents the time required to propagate those packets. When comparing the results achieved with the Z-MAC and secure Z-MAC protocols, it is clear that the secure Z-MAC produces superior results even when the network is inundated with an additional 60,000 packets. Additionally, it can also be observed in Figure 11 that the Z-MAC protocol increased the delay in packet transmission between nodes, resulting in a reduction in transmission time, which implies that packets were substantially lost, and data integrity was not maintained. For instance, as presented in Figure 11, the number of packets increased from 20,000 to 120,000, the transmission time changed abruptly in the case of the Z-MAC protocol, and there were significant delays observed. However, that was not the case with the secure Z-MAC protocol. The transmission time was seen to be linear for all transmitted packets.

However, there was only one instance of delay in the case of the secure Z-MAC, which was caused by the increased number of packets to be encrypted at the start of execution, the weighted moving average for packets, and the change in the trust factor among the nodes. When compared to Z-MAC, secure Z-MAC significantly improves network speed and data integrity; this is most notable in the time required to send various quantities of packets, since packet transmission was uniform, secure, and consistent.

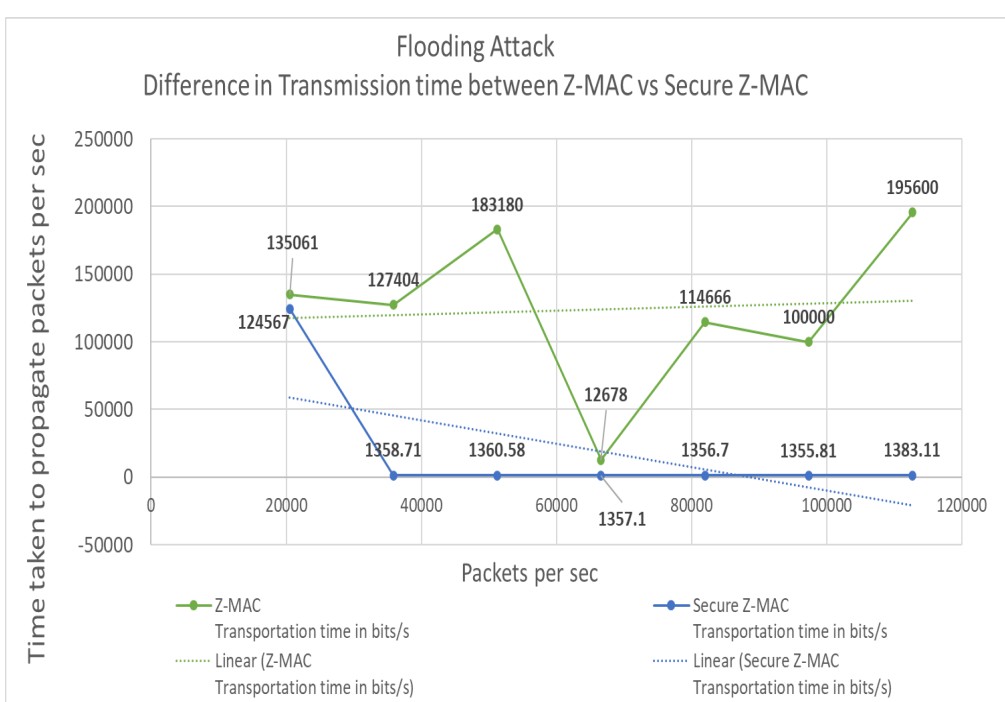

**Figure 11.** Graph simulating the transmission time during a flooding attack against the packets transmitted in byte format in the secure Z-MAC scheme compared to Z-MAC.

### 5.2.5. Blackhole Attack

A blackhole attack was launched in the network with 12,000 packets. The performance of Z-MAC and secure Z-MAC was compared in terms of transmission time, which is represented in the following table.

In Figure 12, it can be observed that when a network is subjected to a blackhole attack, the secure Z-MAC protocol outperforms the standard Z-MAC protocol. This occurred because the Z-MAC protocol is incapable of authenticating the source and destination of message packets. However, when using the secure Z-MAC protocol, such nodes are excluded from all data transmissions, and performance returns to normal. Following the attack, it was observed that the performance of the Z-MAC protocol was degraded in comparison to the secure Z-MAC, as illustrated in Table 8 and Figure 12. Packets were dropped and transmission time was enhanced, resulting in packets being delayed or transmitted to the attacker node. For instance, as presented in Figure 11, the number of packets increased from 20,000 to 120,000, the transmission time changed abruptly in the case of Z-MAC protocol and there were significant delays observed. However, this was not the case with the secure Z-MAC protocol. The transmission time was seen to be linear for all transmitted packets. When a node was deleted from the trusted node list and neighbor discovery, and a new time average was computed, the secure Z-MAC transmission time exhibited some spikes at the start of execution. This delay in time was there to check the trusted node, but the data were not tampered with at all.

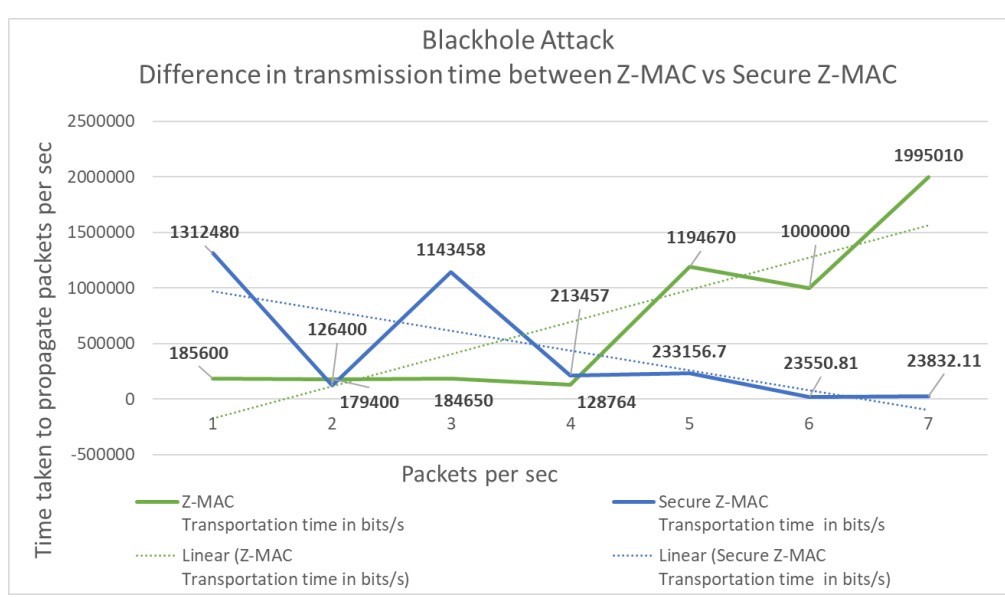

**Figure 12.** Graph simulating the transmission time during a blackhole attack against the packets transmitted in byte format in the secure Z-MAC scheme compared to Z-MAC.

**Table 8.** Total transmission time during a blackhole attack for nodes processed using the secure ZMAC scheme compared to Z-MAC.

| Nodes | Packets/sec in Bytes | Z-MAC Transmission Time in bits/s | Secure Z-MAC Transmission Time in bits/s |
|---|---|---|---|
| 5 | 50480 | 185600 | 1312480 |
| 10 | 65840 | 179400 | 126400 |
| 15 | 81200 | 184650 | 1143458 |
| 20 | 560566 | 128764 | 213457 |
| 25 | 781920 | 1194670 | 233156.7 |
| 30 | 9007280 | 1000000 | 23550.81 |
| 35 | 10012640 | 1995010 | 23832.11 |

### 5.2.6. DDOS Attack

A DDOS attack was launched in the network with 12,000 packets. The performance of Z-MAC and secure Z-MAC was compared in terms of transmission time, which is represented in the Table 9.

**Table 9.** Total transmission time during a DDOS attack for nodes processed using the secure ZMAC scheme compared to Z-MAC.

| Nodes | Packets/sec in Bytes | Z-MAC Transmission Time in bits/s | Secure Z-MAC Transmission Time in bits/s |
|---|---|---|---|
| 5 | 50480 | 115509 | 152461 |
| 10 | 65840 | 139450 | 124000 |
| 15 | 81200 | 144295 | 133690 |
| 20 | 560566 | 161930 | 140442 |
| 25 | 781920 | 194636 | 158101 |
| 30 | 9007280 | 211430 | 153561 |
| 35 | 1001264 | 241873 | 159023 |

In Figure 13, it is observed that when compared to the Z-MAC protocol, the secure Z-MAC protocol is much more efficient. When a DDOS attack occurs in a network, a comparison of Z-MAC and the secure Z-MAC protocol revealed that the secure Z-MAC protocol outperforms the regular Z-MAC protocol. For instance, as presented in Figure 13, a linear change in transmission time was observed compared to the Z-MAC protocol. This

occurred because the Z-MAC protocol lacks a mechanism for authenticating the source and the destination of message packet routes.

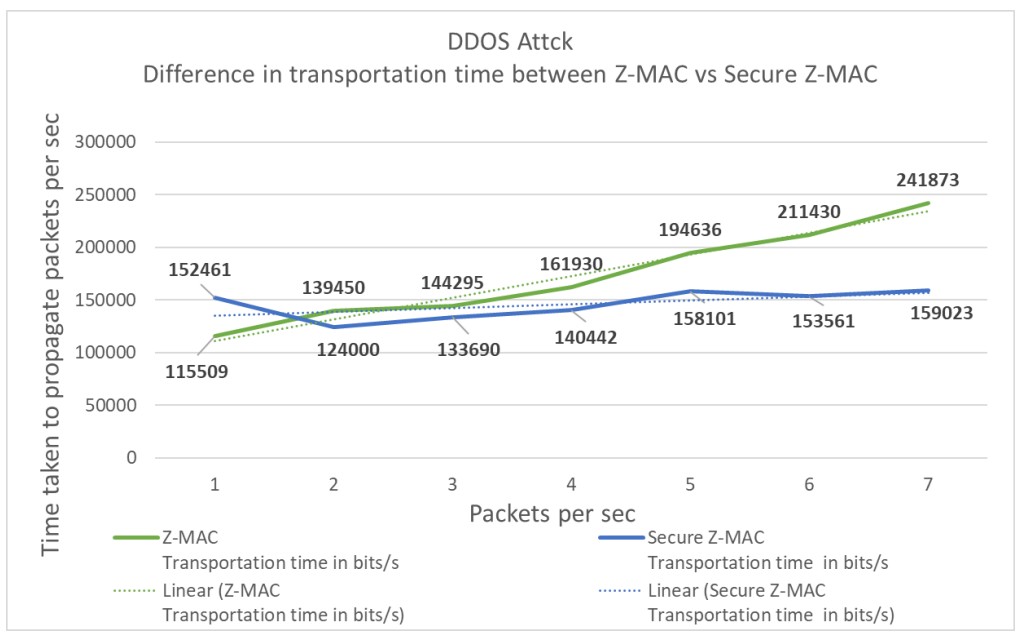

**Figure 13.** Graph simulating the transmission time during a DDOS attack against the packets transmitted in byte format in the secure Z-MAC scheme compared to Z-MAC.

### 5.3. Z-MAC and the Secure Z-MAC Protocol

A comparison of Z-MAC and the secure Z-MAC protocol is carried out in the current section, which focuses on the stages of the protocol as well as simulation results. A comparison of the stages of Z-MAC and the secure Z-MAC protocol is represented in the Figure 14.

| Protocol | Neighbor discovery | Local time Framing(DRAND) | Transmission control(LCL,HCL) | Receiving schedule | Local time synchronization | IHOP | Elliptic curve(Encryption) | Key sharing(Public, Private) | XORING |
|---|---|---|---|---|---|---|---|---|---|
| Z-mac | X | X | X | X | X | | | | |
| Secure Z-MAC | X | X | X | X | X | X | X | X | X |

**Figure 14.** Comparison of the stages in Z-MAC and the secure Z-MAC protocol.

The performance of the protocols at the time of the flooding attack, blackhole attack, and DDOS attack was also compared.

Figure 15 illustrates that when nodes in a WSN are detected and compromised during DDOS, flooding, or blackhole attacks, there is a significant loss of packets. Additionally, there was a significant delay in the transmission time process and low network efficiency, as shown by network disruption, network delay, and the compromised network. However,

in the instance of the secure Z-MAC protocol, when nodes were detected and hacked, only a small number of packets were lost, and those packets were encrypted, enhancing the security of the data being sent. Additionally, there was little delay in the transmission time. Moreover, the adoption of the secure Z-MAC protocol improved network resilience and data integrity under DDOS, blackhole, and flooding attacks. This was due to the network's great efficiency and security, which were achieved via the implementation of the secure Z-MAC protocol.

| (Flooding, DDOS, Blackhole ) Attack. | | | | |
|---|---|---|---|---|
| Protocol | Node | Packets | Transmission Time | Network Efficiency |
| Z-MAC | Identified | Major Lost | Minor Delay | Network Interruption |
| | | | | Network Delayed |
| | Compromised | Minor Lost | Major Delay | Network Reliability |
| | | Encrypted Packets | | Network Compromised |
| | | | | Data Integrity |
| Secure-Z-MAC | Identified | Major Lost | Minor Delay | Network Interruption |
| | | | | Network Delayed |
| | Compromised | Minor Lost | Major Delay | Network Reliability |
| | | Encrypted Packets | | Network Compromised |
| | | | | Data Integrity |

**Figure 15.** Comparison of Z-MAC and the secure Z-MAC protocol in terms of performance during the flooding attack, DDOS attack, and blackhole attack.

The analysis of the protocol was performed by conducting statistical analysis, wherein the primary focus was on the elements, such as size of the packets, number of nodes, and type of attack. Different variations of these elements were considered and tested to ensure that the secure Z-MAC protocol is secure enough to use in a WSN. Also, code analysis helped in developing a code using C++ language, which allowed us to re-run simulations, and the results of the simulations helped in validating the protocol.

## 6. Discussion

From the above-discussed results and analysis, it can be observed that the secure Z-MAC protocol has higher efficiency, security, and productivity compared to the basic Z-MAC protocol. The primary goal of the study was to improve the security of the Z-MAC protocol by making it very safe when used in a wireless network. By designing and implementing a secure Z-MAC protocol, the present study accomplished its goal. Additionally, although the network was subjected to a variety of assaults, including flooding, blackhole, and DDOS attacks, it was determined that the secure Z-MAC protocol secures the network. The newly developed secure Z-MAC protocol has been validated using a variety of attacks. Also, the current research guarantees a significant amount of security analysis

of protocols at a theoretical level, to ensure the security of the implemented protocol. In this article, significant contributions have been made in the field, such as model mining, operations improvement, code generation using code analysis, and statistical analysis. The current research reviewed many code-analysis models to improve the security protocol using C language, to operate and manage any attacks that could have occurred at a source code level.

The current research has made the following additional contributions:

(1) It conducted a security analysis of the secure Z-MAC protocol using code analysis and statistical analysis to ensure that the protocol is valid and secure.

(2) It allowed improvement of the simulations, which helped in making sure that the WSN is secure from the malicious attacks such as flooding attacks, blackhole attack, and DDOS attack.

### 7. Conclusions

The secure Z-MAC protocol was found to be more efficient and secure compared to the basic Z-MAC protocol. Additionally, the implementation of the IHOP mechanism and ECE helped in achieving higher security for the protocol. The secure Z-MAC protocol can be applied in various aspects. For instance, one of the most significant uses of the secure Z-MAC protocol may be the vehicle area network, which aids in boosting vehicular traffic on highways while also improving individual safety and reducing accidents. Additional applications include information transmission inside a cognitive radio network, which would protect the network against eavesdropping and the sharing of erroneous information. The protocol will be evaluated in the future against a variety of network assaults, including a Sybil attack, message-modification attacks, wormhole attacks, and attacks on the system's multiple levels. Additionally, the future plan calls for the development of hardware or software that implements the proposed secure Z-MAC protocol, enabling the safe transmission of essential communications. The hardware would be connected to the network's user devices through an application, ensuring a secure network connection.

**Author Contributions:** M.N.A. Writing-Original, reviewing, Editing, & design the work, A.A.E. and A.A.M.M. supervision, Data curation & reviewing. All authors have read and agreed to the published version of the manuscript.

**Funding:** This research received no external funding.

**Institutional Review Board Statement:** Not applicable.

**Informed Consent Statement:** Not applicable.

**Data Availability Statement:** No new data were created or analyzed in this study. Data sharing is not applicable to this article.

**Conflicts of Interest:** The authors declare no conflict of interest.

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
