# Peer review of "Secure Z-MAC Protocol as a Proposed Solution for Improving Security in WSNs"

_information, doi:10.3390/info13030105_

Round 1
Reviewer 1 Report
- The Abstract of the manuscript-at-hand appears shallow and needs to be documented in a categorical manner, i.e., it needs to delineate the significance of the research-at-hand succinctly by primarily elaborating on the overall domain, illustrating the underlying challenges of the domain, and summarizing the key challenges especially addressed by this particular manuscript. The same is also true for the Introduction section.
- The contributions of the manuscript-at-hand need to be documented at the end of the Introduction preferably in a serialized manner, (1)__; (2)__; and (3)__.
- The notion, Foreign Attacks and Interior Attacks, should be replaced with External Attacks and Internal Attacks throughout the manuscript.
- While illustrating the Types of Security Attacks in Networks (Section 2.1), the authors should also highlight the numerous technological variants, i.e., with appropriate references, that could be employed in order to strengthen the resilience of the sensor networks (cryptography, trust management, and blockchain). The authors can refer to the following research literature in this regard:
- Employing Blockchain Technology to Strengthen Security of Wireless Sensor Networks. IEEE Access, vol. 9, pp. 72326-72341, 2021 (10.1109/ACCESS.2021.3079708).
- Trust Computational Heuristic for Social Internet of Things: A Machine Learning-based Approach. ICC 2020 - IEEE International Conference on Communications (10.1109/ICC40277.2020.9148767).
- Cryptography Methods for Software-Defined Wireless Sensor Networks. IEEE 27th International Symposium on Industrial Electronics (ISIE), 2018 (10.1109/ISIE.2018.8433630).
- A comparison of the existing state-of-the-art in terms of their pros and cons is also indispensable in the section, Literature Review.
- Section 3.1, Proposed Solution: Secure Z-MAC, should be a completely separate section and not a part of the Results.
- A critical analysis of the Experimental Results, i.e., especially for Figure 6 - 12 is also indispensable here.
- There are some issues with grammar, jargon, and sentence structure and careful proofreading would be appreciated.
Reviewer 2 Report
The paper proposes the Z-MAC protocol as a solution for security in WSNs.
The paper is well organized. The state of the art is also good. The protocol is well described.
However, the authors do not prove the security of the protocol by using a recognized analysis method. This raises many doubts about the validity of the whole work.
Reviewer 3 Report
The proposal is sound. The descriptions are exhaustive and the results seem coherent and correct. In general I think the paper may be accepted, but some minor changes are recommended:
- Some additional mathematical expressions are needed. Algorithms 1 and 2 are fine, but their mathematical foundations should be described with more details
- Figures are not well presented. Color, tags, data, etc. should be reorganized in a more coherent way. Now, Figures are very hard to read and understand
Round 2
Reviewer 1 Report
Thank you for addressing the comments which have considerably improved the quality of the manuscript-at-hand.
Author Response
Thank you for the comments and suggestions, Appreciated.
This manuscript is a resubmission of an earlier submission. The following is a list of the peer review reports and author responses from that submission.